# Gazing through time and beyond the health sector: Insights from a system dynamics model of cardiovascular disease in Australia

Cindy Q. Peng[1]*, Kenny D. Lawson[2,3], Mark Heffernan[2,4], Geoff McDonnell[5], Danny Liew[6], Sean Lybrand[7], Sallie-Anne Pearson[8], Henry Cutler[9], Leonard Kritharides[10], Kathy Trieu[11], Quan Huynh[12], Tim Usherwood[11,13], Jo-An Occhipinti[1,3,14]

1 Decision Analytics, The SAX Institute, Sydney, Australia, 2 Adjunct, Western Sydney University, Sydney, Australia, 3 Brain and Mind Centre, University of Sydney, Sydney, Australia, 4 Dynamic Operations, Sydney, Australia, 5 Adaptive Care Systems, Sydney, Australia, 6 School of Public Health and Preventive Medicine, Monash University, Melbourne, Australia, 7 Amgen Europe GmbH, Rotkreuz, Switzerland, 8 Centre for Big Data Research in Health, University of New South Wales, Sydney, Australia, 9 Centre for the Health Economy, Macquarie University, Sydney, Australia, 10 Concord Repatriation General Hospital, University of Sydney, Sydney, Australia, 11 The George Institute for Global Health, University of New South Wales, Sydney, Australia, 12 Baker Heart and Diabetes Institute, Melbourne, Australia, 13 Westmead Applied Research Centre, University of Sydney, Sydney, Australia, 14 Computer Simulation & Advanced Research Technology (CSART), Sydney, Australia

* cindy.qc.peng@gmail.com

## Abstract

### Objective

To construct a whole-of-system model to inform strategies that reduce the burden of cardiovascular disease (CVD) in Australia.

### Methods

A system dynamics model was developed with a multidisciplinary modelling consortium. The model population comprised Australians aged 40 years and over, and the scope encompassed acute and chronic CVD as well as primary and secondary prevention. Health outcomes were CVD-related deaths and hospitalisations, and economic outcomes were the net benefit from both the healthcare system and societal perspectives. The eight strategies broadly included creating social and physical environments supportive of a healthy lifestyle, increasing the use of preventive treatments, and improving systems response to acute CVD events. The effects of strategies were estimated as relative differences to the business-as-usual between 2019–2039. Probabilistic sensitivity analysis produced uncertainty intervals of interquartile ranges (IQR).

### Findings

The greatest reduction in CVD-related deaths was seen in strategies that improve systems response to acute CVD events (8.9%, IQR: 7.7–10.2%), yet they resulted in an increase in CVD-related hospitalisations due to future recurrent admissions (1.6%, IQR: 0.1–2.3%).

**Data Availability Statement:** All relevant data are within the paper and its Supporting Information files.

**Funding:** This work is funded by Amgen Australia. The funder has no role in the design of the study and analysis and interpretation of data. Co-authors CP and JO received salary from the Sax Institute, partially funded by this work. KL, MH, and GM received consultancy fees from the project funding. Co-author SL is an employee of Amgen and participated as an individual member (see Author's contributions).

**Competing interests:** Mr. Lybrand reports salary and benefits from Amgen Australia Pty Ltd, the funder of this work. A/Professor Jo-An Occhipinti reports a grant from Amgen Australia to undertake the research; and A/Professor Jo-An Occhipinti, Ms Cindy Peng, A/Professor Kenny Lawson, Professor Mark Heffernan, and Dr Geoff McDonnell report receiving funding from Amgen Australia to undertake there search. A/Professor Jo-An Occhipinti is Managing Director of Computer Simulation & Advanced Research Technologies (CSART), an international not-for-profit organization building infrastructure and capacity in the use of systems modelling and simulation to inform health and social policy. Danny Liew has received consulting fees from Abbvie, Astellas, AstraZeneca, Bristol-MyersSquibb, Novartis, Pfizer, Sanofi and Edwards Life sciences, but not related to the present study. Danny Liew has been awarded research grants from Abbvie, Astellas, Amgen, AstraZeneca,Bristol-Myers Squibb, Pfizer and Sanofi, but not related to the present study. Mark Heffernan is the CEO of Dynamic operations, one of the Australian distributors of Stella Architect, the modelling software used in this study. This does not alter our adherence to PLOS ONE policies on sharing data and materials.

This flow-on effect highlighted the importance of addressing underlying CVD risks. On the other hand, strategies targeting the broad environment that supports a healthy lifestyle were effective in reducing both hospitalisations (7.1%; IQR: 5.0–9.5%) and deaths (8.1% reduction; IQR: 7.1–8.9%). They also produced an economic net benefit of AU$43.3 billion (IQR: 37.7–48.7) using a societal perspective, largely driven by productivity gains. Overall, strategic planning to reduce the burden of CVD should consider the varying effects of strategies over time and beyond the health sector.

## Introduction

Despite a dramatic decline in death rate of cardiovascular disease (CVD) over the last few decades, CVD remains the leading cause of death in Australia and the rate of decline has slowed [1]. This means not only loss of health to individuals but also an economic burden to nations as a result of rising healthcare costs and lower productivity [2–4]. In addition, an ageing population and high prevalence of risk factors have led scientific and clinical communities to caution a growing burden of CVD in the future [5]. Policy decisions aiming to slow down or reverse the growth of the CVD burden are challenging given the complexity of CVD. They need to balance prevention and treatment, clinical and behavioural influences, and individual interventions and population-wide changes. Population health strategies aiming to reduce the burden of CVD may need to include multiple system-wide strategies to address these challenges.

Simulation models provide a tool for policy analysis by incorporating evidence from different sources and different domains to represent the real world and forecasting population-wide effects from strategies [6]. They can identify where costs and benefits lie, and allow population health strategies to be tested in a safe virtual environment before being implemented [6]. Existing policy models for CVD have been developed for specific countries and have mostly focussed on coronary heart disease [7–12]. A comprehensive Australian study completed in 2010 estimated the cost effectiveness of a wide range of interventions for chronic conditions, mainly derived from Markov models. Given the complexity of CVD, a contemporary systems-level model may be necessary to account for the dynamic influences between primary and secondary prevention, competing risks, and the interactions between interventions.

System dynamics (SD) modelling is an approach to policy simulation founded on two key premises. Firstly, systems can be complex, meaning nonlinear relationships between components of the systems result in behaviours such as feedback loops and long delays between cause and effect, and these behaviours can change dynamically [13]. Secondly, SD modelling uses a participatory process to elicit qualitative understandings from key actors [14], and quantifies these understandings with data and research evidence. It is therefore uniquely placed to study complex systems, which are often responsible for problems that resist easy solutions, and to build a common understanding and consensus for actions. SD models of CVD have previously compared interventions ranging from regulations, clinical management, behavioural support, and health promotion and access, and found that the best option depended on the time horizons used and the outcomes of interest [10–12].

The purpose of the present study was to construct a whole-of-system model to inform public health strategies to reduce the burden of CVD in Australia. The aims of this paper are twofold: to describe the development of the SD model, and to demonstrate its use to inform strategic priority setting on reducing the health and economic burden of CVD.

## Materials and methods

### Participatory model development process

A SD model was developed through four stages: 1) participatory systems mapping and conceptual diagram development; 2) conversion of the conceptual diagram to a computational model via parameterisation with numeric inputs; 3) design, integration, and testing of strategies; and 4) model validation and uncertainty analysis. A modelling consortium of 23 multidisciplinary members was established to inform all four stages of model development via group model-building workshops and out-of-session individual consultations. They included experts in clinical management, public health research, consumer groups, health economics, industry, and policy agency representatives. The participatory modelling approach employed has been described in detail elsewhere [15]. Model construction and analysis were performed using Stella Architect ver. 1.9.2 (www.iseesystems.com).

An initial group model-building workshop was conducted to develop a conceptual diagram. The modelling consortium was guided to discuss and map relevant systems and factors contributing to the burden of CVD. The project team reviewed audio recordings and systems maps after the workshop, and identified patterns of interest and recurring themes [14]: 1): in addition to acute deaths from CVD, there was a growing recognition on chronic burden; 2) reducing the burden of CVD would require both wider use of clinical treatments and lifestyle changes.

A modified Delphi method was used to reach consensus on the prioritisation and design of strategies [14]. Parallel to conceptual model development, an anonymous survey was distributed to the modelling consortium. It included 16 potential strategies supported by literature review and options to propose additional strategies, and the modelling consortium was asked to select eight strategies that are likely to be effective in Australia. An aggregated result was reported back to the modelling consortium in the first workshop to encourage group discussion on the rationale and whether the list needed to be modified. The result of this discussion was incorporated into a preliminary version of the model, which acted as the basis for a second workshop discussion with a focus on feasibility and implementation of the modelled strategies. A summary report confirming the results of discussions was circulated afterwards, as an opportunity for endorsements or final individual disagreement. This process led to a consensus to include a wide range of strategies that improve both clinical management and the broader environment. It also highlighted difficulty in narrowing down implementation mechanisms of strategies due to lack of high-quality evidence, and both future research and models should address this gap.

### Model structure and numeric inputs

A SD model is a compartmental model primarily accounting for numbers of people in different states, called "stocks" (e.g. people living with CVD), and the numbers transitioning between the states, called "flows" (e.g. annual hospitalisations due to CVD). For the CVD model, numeric inputs of the rate of flow (i.e. probabilities of transitions) and the values of stocks at the start of the time horizon were based on existing data, derived from evidence identified through a literature review and assessed using the GRADE system [16]. The model simulated values for the rest of the time horizon. A detailed description of key assumptions, model equations, and numeric inputs are provided in **S1 Appendices**.

The modelled population comprised Australians aged over 40 years, including changes in the size and age structure over time. The population entered the model through ageing and migration and exited through deaths due to CVD and non-CVD reasons (to account for

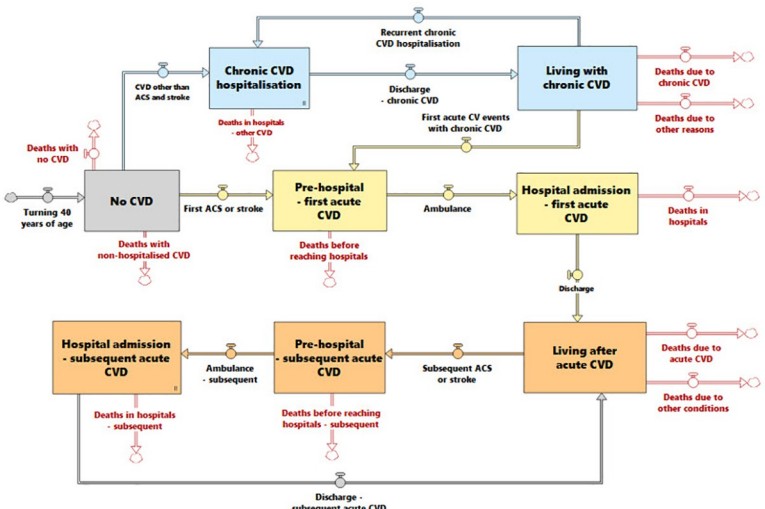

**Fig 1. Simplified graphic representation of the model structure.** ACS = acute coronary syndrome, CVD = cardiovascular disease.

competing risks of deaths from non-CVD causes). Medium demographic rate projections were assumed, consistent with 'Series B' projections published by the Australian Bureau of Statistics (ABS) [17].

Primary outcomes of interest were hospitalisations and deaths from CVD, and secondary outcomes were economic net benefits of strategies. The model time horizon was 20 years from intervention implementation and the outcomes were examined at 3, 10, and 20 years. This timeframe was chosen for pragmatic reasons; policy decisions are often made with consideration of expected outcomes at 3 (election cycles) and 10 years (long term planning frameworks) [18, 19], and the 20-year time horizon allowed longer term effects of strategies to be examined. The core module of the model accounting for the number of people hospitalised for and dying from CVD is shown in a simplified graphic representation (Fig 1). It comprised acute CVD and chronic CVD. Acute CVD events included acute coronary syndromes (ACS) and stroke and chronic CVD included all other CVD, as defined by the 10th revision of the International Statistical Classification of Diseases and Related Health Problems (ICD-10). This practical categorisation was consistent with data sources. Each acute CVD event included pre-hospital and hospital stages, and chronic CVD considered hospital and post-hospital-discharge. They were also separated into initial and recurrent events/hospitalisations. The model was structured to account for markedly different probabilities of deaths, probabilities of transitions to other states, and lengths of time in each state. Probabilities were derived from data from cohort studies and registries, and were calibrated against cause of deaths recorded on the ABS and primary diagnosis recorded on the National Hospital Morbidity Database (NHMD) [20].

## Strategies and scenarios

The eight strategies are described in Table 1, and more details are included in **S1 Appendices**. They fall into three categories:

1. Creating an environment supportive of healthy lifestyle: salt reduction in processed food, reducing sedentary behaviours, and reducing smoking prevalence. These lowered the incidence of the initial and recurrent CVD in a broad population

**Table 1. Summary of strategies included.**

| Strategy | Description | References |
|---|---|---|
| **Supportive environment** | | |
| **Salt reduction in processed food** | Voluntary reformulation of processed food reduces their sodium content by 10% over 4 years, and subsequently population-level blood pressure levels and CVD risk. | [21–27] |
| **Improving physical activity** | Building physical environment supportive of active lifestyle and increasing awareness through social marketing campaigns reduces the proportion of inactive adults by 5% each, and associated CVD risks. | [28–30] |
| **Reducing smoking prevalence** | Intensified anti-smoking efforts, including increasing prices, standardising pack sizes, and continuous social marketing campaigns, further drive down the smoking prevalence to approximately 6% in 20 years. | [31, 32] |
| **Response to acute CVD** | | |
| **National public access defibrillator program** | An optimally implemented national public access defibrillator program can reduce chances of deaths after an out-of-hospital cardiac arrest by 50%. | [33–36] |
| **Reducing pre-hospital delay** | Reducing pre-hospital delay to a target level of 2 hours lowers chances of deaths after acute CVD by 20%. | [37] |
| **Preventive medications and treatments** | | |
| **Prescription of preventive medications** | Raising public awareness and supporting primary care physicians can increase the proportion of eligible adults (as indicated by absolute CVD risk and previous CVD) that receive blood-pressure and lipid lowering medications by 5 to 10% | [38–40] |
| **Adherence to preventive medications** | Raising public awareness and supporting primary care physicians can increase adherence to blood-pressure and lipid lowering medications by 10 to 20% | [38–40] |
| **Cardiac rehabilitation program** | Increasing referral, attendance, and completion of cardiac rehabilitation programs by 15 to 30% lowers risks of recurrent myocardial infarction. | [41–43] |

2. Increasing the use of preventive medications and treatments: increasing medication prescription, medication adherence, and use of cardiac rehabilitation programs. These lowered incidence of the initial and recurrent CVD in those who are eligible.

3. Improving systems response to acute CVD events: a national public defibrillator program and reducing pre-hospital delay. These lowered death rates from acute CVD.

Direct impacts of strategies on individuals' risks of CVD incidence and death were based on estimates from randomised controlled trials and large cohort studies (see Table 1). The population-level effects of strategies were dependent on their implementation mechanisms, for example, roll-out and uptake. The assumed consequences of the strategies were based on outcomes from previous similar programs where available, or most plausible estimates of the modelling consortium when direct evidence was unavailable, with uncertainty explored in the probabilistic sensitivity analysis. Given limited high-quality evidence in implementation mechanisms and the purpose of the model as a strategic rather than operational decision support tool, the strategies were defined broadly rather than specific public health interventions.

Model insights were derived from comparing various scenarios to the base-case, i.e. projection using the current age-specific incidence rates and demographic changes over the model time horizon. Scenarios tested included implementation of individual strategies and combinations of strategies, as well as strategies under other possible implementation conditions (e.g. different coverage, different timeframes).

Economic impacts of strategies were assessed through assigning unit values to the consequences of strategies, incorporating both health system and societal perspectives. Under the health system perspective, net benefit was calculated as monetised health gain less health service cost from CVD (e.g. general practitioner consultations, medications, hospitalisation, and rehabilitation). Health gain was monetised as Quality Adjusted Life Years (QALYs) gained multiplied by AU$50,000, an amount broadly considered to be a threshold willingness to pay for 1 QALY in the health sector [44]. Health service cost was stratified by payers, i.e. the Commonwealth Government of Australia, the state and territory governments, and consumers. Under the societal perspective, net benefit was the sum of health service cost, household cost (out-of-pocket health service cost and carer time), productivity gains (frictional-cost method), and monetised health gains (i.e. AU$50,000 multiplied by QALYs gained) [44]. The health system cost of managing populations free of, or living with, CVD was also estimated for completeness. This is important when considering the full impacts of strategies to prevent CVD and extend life expectancy. All costs and outcomes were discounted at 5% per annum, consistent with common practice of economic evaluation in the health sector [45].

The cost of implementing strategies were not included, as the specific interventions under respective strategies were not defined. The costs (or costs avoided) of implementing strategies were only reflected in flow-on health service costs, for example, changes in utilisation of general practitioner consultations, medications, hospitalisations, and rehabilitation services. The purpose of the exposition is to first identify the flow-on economic effects of introducing strategies, which then informs the prioritisation of developing and evaluating specific interventions that may deliver such effects. To accommodate the strategic intent of the model to influence policy from multiple sectors, the economic approach was intentionally designed to be flexible to tailor the method to suit particular decision-maker needs, if needed. This in-built flexibility includes, for instance, an ability to vary discount rate(s), valuation of life (e.g. value of statistical life), and valuation of productivity (e.g. human capital approach). This approach prepared the policy model to inform the development and evaluation of both health sector initiatives and whole-of-government initiatives and align with different guidance as appropriate (e.g. health sector, Treasury).

## Model calibration, validation, and uncertainty analysis

Graphic representations of model equations underwent multiple qualitative logic checks with the modelling consortium for structural verification. Dimensional consistency and extreme conditions were also tested to ensure structural validity. Key health outcomes, including hospitalisations and deaths from CVD in Australia, were used for calibration against administrative data recorded on ABS and NHMD from 2011–17. Parameter values were verified against published evidence and data, or consensus within the modelling consortium in the absence of directly relevant data.

Uncertainty in parameter values of model variables was explored in probabilistic sensitivity analysis generating uncertainty measures of interquartile range (IQR) in model outcomes. Selected model variables included direct impacts and implementation scenarios of each strategy. Distributions and uncertainty ranges of parameter values were sourced from the literature, and where evidence was limited, uniform distributions of a broad range of values were assumed (see **Appendix 1 in S1 Appendices** for details of variables selected, uncertainty distribution, and outputs). Latin hypercube sampling was used and 250 sets of values within the distributions of all selected variables were considered sufficient to derive uncertainty intervals [46].

## Results

The model reproduced the historical data pattern of CVD deaths and hospitalisations from 2011 to 2018 in Australia. Under the base-case scenario, it forecast an 80% increase in annual deaths and a 77% increase in annual hospitalisation from CVD from 2019 to 2039, equating to a total of approximately 1.3 million deaths and 11.7 million hospitalisations caused by CVD over 20 years in Australians over 40 years old. The model estimated an approximate distribution of 55:20:25 between deaths occurring during the pre-hospital stage of acute events, in-hospital stage of acute events, and in the community, while chronic CVD accounted for 60% of CVD-related hospitalisations.

Health outcomes of the three groups of strategies and their IQR at 3 years, 10 years, and 20 years are shown in Figs 2 and 3. The relative reduction of hospitalisations and deaths compared to the base-case at 20 years was 4–5 times higher than that at 3 years, reflecting the delayed onset of benefits from strategies. Over the model horizon of 20 years, the strategies targeting the pre-hospital stage of acute CVD events (i.e. improving cardiac arrest survival and reducing pre-hospital delay) were forecasted to prevent 8.9% (IQR: 7.7–10.2%) of CVD-related deaths compared to the base-case. However, they resulted in an increase of CVD-related hospitalisations by 1.6% (IQR: 0.1–2.3%). Strategies targeting a supportive environment, on the other hand, achieved significant reductions in both hospitalisations (7.1% reduction; IQR: 5.0–9.5%) and deaths (8.1% reduction; IQR: 7.1–8.9%).

Economic outcomes of the three types of strategies and their IQRs at 3 years, 10 years, and 20 years are shown in Table 2. Strategies targeting supportive environment generated greatest net benefits under both the health-sector perspective (AU$12.9 billion; IQR: $5.1–20.7 billion) and the societal perspective (AU$34.2 billion; 95% uncertainty interval: $13.4–55.4 billion) over 20 years. The economic net benefits of medications and treatments under societal perspective were negative at 3 years but grew to AU$21.7 billion at 20 years, 88% of which are attributed to productivity gains. The direct health costs incurred were approximately AU$4.91 billion, AU$3.21 billion, and–AU$2.91 billion (cost-saving) respectively for strategies targeting medication and treatment, pre-hospital response, and supportive environment.

## Discussion

The present paper introduces a novel SD model to inform CVD policymaking in Australia, including the process of building the model, and a high-level demonstration of how the model can be used to inform strategic priority setting.

The model demonstrated that the burden from CVD would grow significantly in the next few decades under the base case scenario, highlighting the consequences of a lack of additional actions. Strategies that create an environment supportive of healthy lifestyle were forecasted to deliver the greatest benefits, yet they cannot be implemented by the health sector alone. Moreover, the payers and beneficiaries of public health strategies often reside in different levels of government and different sectors of the economy. For example, the cost of preventive medications and treatments are mostly borne by the health sector of the Commonwealth Government of Australia (e.g., subsidised via Medicare Benefits Schedule and Pharmaceutical Benefits Schedule), while the cost saved from averted public hospitalisations occurs at both the Commonwealth and state and territory government levels, and significant productivity gains benefit the whole society. In summary, the model showed the large degree of potential health and economic benefits that could be achieved with collaboration between different levels of governments and different sectors of the society as well as consequences if no different action is taken, and thereby building a strong case for broader and bolder collaboration.

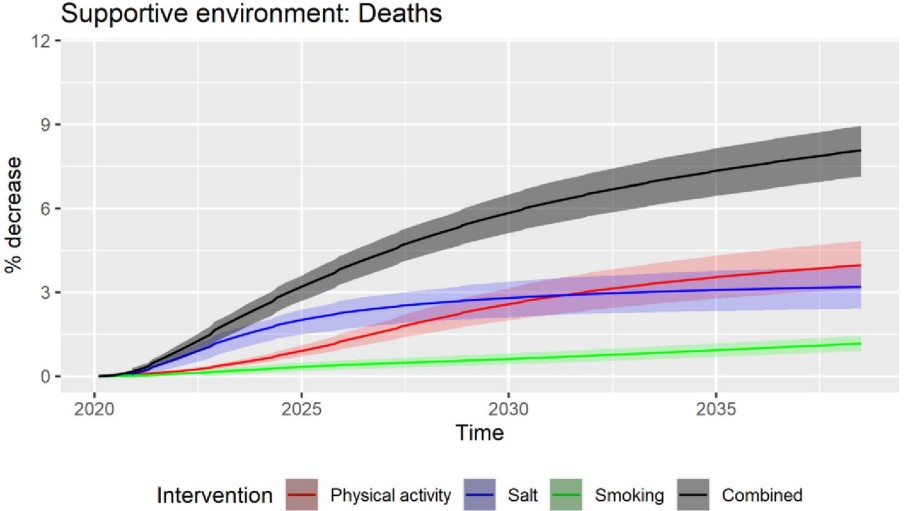

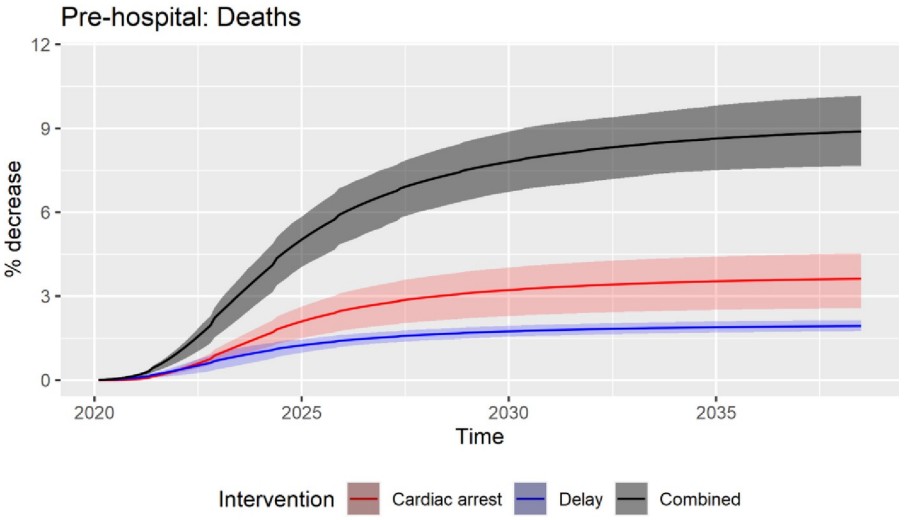

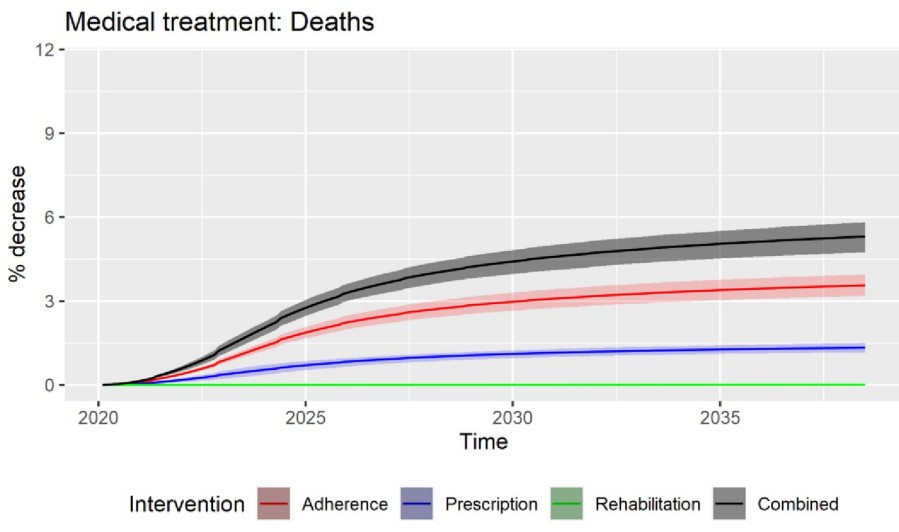

**Fig 2. Relative reduction (%) in cumulative impacts on deaths as a result of strategies and their interquartile ranges at 3, 10, 20 years; strategies are combined into three groups: Supportive environment, pre-hospital, and medication and treatment.**

The economic net benefits not only inform priority setting for government but also provide a reference point for the level of early investments proportional to the size of opportunity. For example, given a cumulative net benefit of over $40 billion under the societal perspective for strategies targeting supportive environment, funding levels for small pilot programs and research projects to explore what works should be assessed with this scale in mind. On the other hand, the direct health cost used in this study, calculated as flow-on effects from implementing the strategies (e.g. costs of medications and hospitalisations across the study population), were small relative to monetised QALY gains, and even smaller when productivity gains are counted within the societal perspective. We intentionally took a broad perspective in valuing human health, but audience with a perspective of direct budgetary impacts may need to note this when considering implications of the findings.

The model captured interrelationships and dynamic changes and anticipated flow-on effects of strategies in the broader systems. Improving response to acute events was the most effective in preventing CVD- related deaths, yet doing so without addressing underlying CVD risk would lead to an increase in hospitalisations and associated costs as survivors undergo recurrent hospitalisations. Homer and Hirsch previously studied the dynamics of 'upstream' and 'downstream' health activities in chronic disease prevention and management and demonstrated a similar feedback principle [13]. Longer life achieved through downstream activities results in more resources immediately required for downstream management, which inadvertently diverts resources from upstream prevention and in turn results in more chronic disease that require downstream management. While it is a moral imperative to save lives in acute incidents, it is important to be aware of the flow-on consequences and the importance of investment in prevention for sustained health of the population.

The effects from the proposed strategies accumulated in a non-linear fashion. This resulted from the interaction of a combination of factors, including implementation parameters (time to fully implement a strategy), biological mechanisms (some strategies have immediate impacts on CVD risks while others take time for the CVD risk to decline), an open population (i.e. accounting for changes in births, deaths and migrations over time), and the probabilities of developing CVD (benefits take time to manifest especially for younger age groups). It is important for decision makers, who often operate in short funding cycles, to be aware of such non-linearities so that strategies that deliver long-term benefits can be considered and outcomes are evaluated within an appropriate timeframe.

Given the dynamic relationship between disease incidence and deaths from CVD, the present study included a diverse mix of strategies targeting different mechanisms. To our knowledge, there is no other modelling study directly comparable under the Australian policy setting. Cobiac et al. developed an important model to assess the potential cost effectiveness of primary prevention interventions from a health sector perspective [47, 48]. This was a Markov model, in discrete time, with a closed cohort, and compared a range of lifestyle, pharmaceutical, and population-wide interventions [47, 48]. The present model showed similar findings in the benefits in salt reduction in processed food and appropriate clinical management, and further demonstrated their significant economic benefits beyond the health sector.

With regards to the future burden of disease from CVD, the AIHW's projection of Australian health expenditure conducted in 2008 assumed a decline of CVD incidence by 40% between 2003 to 2033, given a dramatic decline seen in the prior decades [5]. The present study have projected future CVD burden using age-specific CVD incidence rates observed

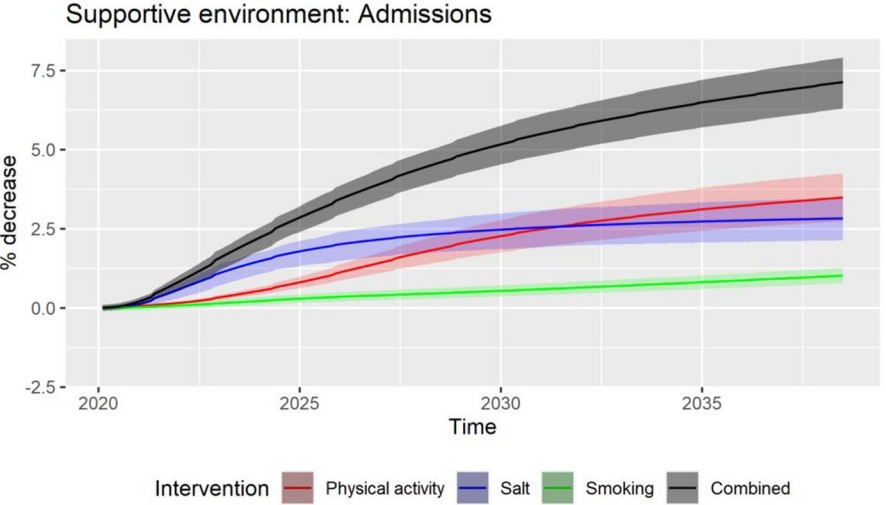

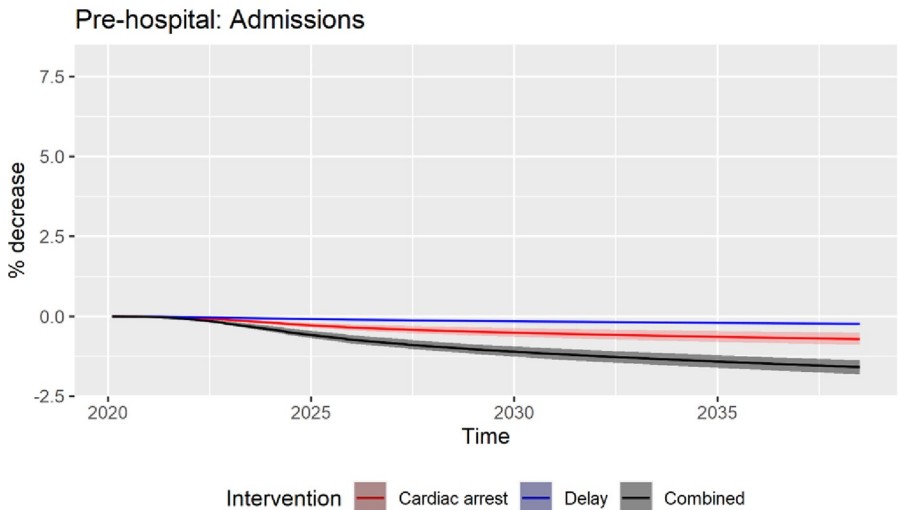

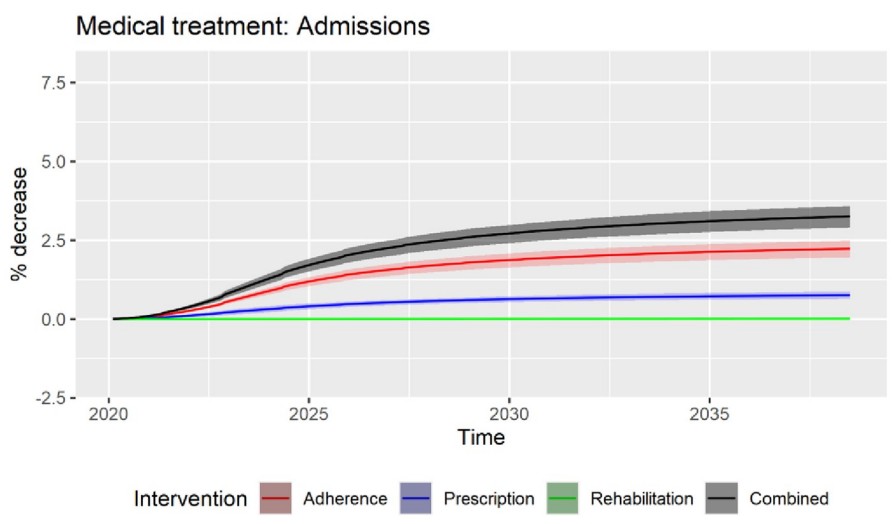

**Fig 3. Relative reduction (%) in cumulative impacts on hospitalisations as a result of strategies and their interquartile ranges at 3, 10, 20 years; strategies are combined into three groups: Supportive environment, pre-hospital, and medication and treatment.**

between 2011–17, as the decline seen in earlier years has plateaued in recent years [20]. Although using an alternative assumption of declining age-specific CVD incidence would result in a lower estimate of hospitalisations and deaths in the base-case, relative impacts of strategies are unlikely to change as they are compared to the same reference point. Therefore, the future CVD burden projected by the present model should be interpreted within the context of comparing impacts of strategies rather than being reported in isolation.

Our participatory model development process is similar to the structure and scripts of group model building (GMB) described by George Richardson and David Andersen in their seminal works in the 1990s [49]. Different from the conventional GMB practice, where the model boundary and structure are almost solely driven by the participants and often rich in stories and feedbacks, the present process was heavily influenced by the practice of evidence-based medicine. This means the model structure and parameter values were suggested by the modelling consortium but were also cross-validated with evidence and data. This is the reason why the model structure is largely compartmental based on biological mechanisms and public health data, with limited feedback loops. This is in line with the strategic purpose and broad scope of the model, and other CVD models using SD methodology reached similar structures [11, 12, 50]. However, future models that investigate the pathways to deliver the strategic priorities should aim to explore feedback dynamics so the specific programs and interventions can be effectively delivered. We have also found that the modelling consortium, comprised of domain experts, had great knowledge and intuitions in optimising locally within their domains. On the other hand, with such broad scope, there were gaps in the collective knowledge of the modelling consortium that needed to be filled with published literature. For future participatory modelling that aims to design specific programs and interventions, we recommend selecting modelling consortium members with diverse yet overlapping expertise, so different views on similar topics can be compared to construct a rich and comprehensive picture. The model has several strengths as a tool to support policy decision-making. The model-building process enabled qualitative mental models from a range of experts, and quantitative data from various sources, to be synthesised, validated and calibrated into one comprehensive model. The SD methodology reflects how complex systems work in the real world and identifies the dynamics that may not be unearthed by traditional analytical methods. The current model has an open population calibrated to the Australian general population, allowing for

**Table 2. Economic outcomes of three types of strategies and their interquartile ranges at 3, 10, and 20 years.**

| Strategies[1] | Net benefit—health systems (AU$ Billion) [2] | | | Net benefit–broader society (AU$ Billion) [3] | | |
|---|---|---|---|---|---|---|
| | 3 years | 10 years | 20 years | 3 years | 10 years | 20 years |
| Supportive environment | 0.36 (0.25, 0.42) | 5.54 (4.72, 6.13) | 17.97 (15.55, 19.82) | 0.61 (0.36, 0.86) | 12.58 (10.81, 14.26) | 43.28 (37.74, 48.70) |
| Pre-hospital | 0.44 (0.34, 0.54) | 3.66 (3.13, 4.13) | 12.17 (10.52, 13.88) | 0.05 (0.04, 0.07) | 3.03 (2.49, 3.50) | 11.47 (9.81, 12.94) |
| Medication and treatment | - 0.10 (-0.20, 0.01) | - 0.77 (-1.1, -0.42) | 3.24 (2.51, 3.86) | - 0.37 (-0.41, -0.31) | 2.24 (1.59, 2.85) | 14.20 (11.75, 16.49) |

[1] The model assumes various duration for strategies to reach their maximum impacts, ranging from 2 to 10 years.

[2] The net benefit under the health sector perspective includes flow-on health service and medication costs and monetised Quality Adjusted Life Years; it does not include cost involved in implementing specific interventions.

[3] The net benefit under the societal perspective includes flow-on health service and medication costs, monetised Quality Adjusted Life Years, other households' costs, and value of productivity; it does not include cost involved in implementing specific interventions.

consideration of competing risks of death from other causes and more accurate attribution of deaths from CVD. It has a wide scope to include acute CVD and chronic CVD, primary and secondary prevention, and a diverse range of strategies encompassing environmental and clinical changes. Finally, the model incorporated the flexibility to adopt multiple economic perspectives such as health systems, society, and different levels of government. This is intended to allow tailored model outputs to different decision makers and stakeholders, and quantify opportunities for closer policy coordination.

The model has several limitations. First, results should be interpreted with the consideration of the uncertainties in outputs propagated from uncertainties in the model structure and input data. Despite efforts to reduce the uncertainties through the participatory process and the triangulation between multiple data sources, they are unlikely to have been eliminated, given that a model is intrinsically a simplification of the real world and that quality and quantity of data are invariably imperfect. In addition, the definitions of acute CVD and chronic CVD were aligned with national datasets, which may be different from those used at the clinical level. Another limitation is that the model currently does not have stratifications of social determinants of health, such as socio-economic status, or Indigenous status, and thus effects of strategies in these subpopulations are not captured in the current iteration. Finally, as available evidence to inform implementation estimates were limited and inconsistent, the current model did not contain detailed implementation scenarios and associated costs, and therefore could not be directly used to make specific operational and investment decisions. For example, it shows the costs and consequences from a plausible degree of improvement in the percentage of high-risk patients who receive preventive medications and how the outcomes compare to those from other strategies, but it does not detail how an initiative could be designed to achieve such degree of improvement. The aim of the model is to inform areas with high potential benefits at a strategic level and directs attention to define which specific interventions can best be developed, implemented, and evaluated. Future iterations of the SD model can then be developed as the new evidence emerge.

## Conclusion

The SD model demonstrated the importance of developing policies to balance different strategic areas, in particular upstream prevention and improving survival after acute CVD. In addition, the non-linear nature of how effects accumulate had important implications in the timeframe of both policy planning and evaluation. The model also built a compelling case for collaboration within and beyond the health sector by quantifying its economic value and highlighting its critical role in reducing the future burden of CVD in Australia. Future iterations of the model can simulate specific interventions with emerging evidence in implementation practice.

## Supporting information

**S1 Appendices. Numeric inputs, key assumptions, and equation list.**
(DOCX)

## Acknowledgments

In addition to the author group, the work was made possible by the generous contributions of expert knowledge from Dr Andrea Schaffer, University of New South Wales; Prof. David Brieger, University of Sydney; Prof. David Roder, University of South Australia; Prof Andrew Page, University of Western Sydney; and Ms Franca Facci, Illawarra Shoalhaven Local Health District.

## Author Contributions

**Conceptualization:** Sean Lybrand, Jo-An Occhipinti.

**Data curation:** Cindy Q. Peng, Kenny D. Lawson.

**Formal analysis:** Cindy Q. Peng, Kenny D. Lawson, Mark Heffernan, Geoff McDonnell, Danny Liew, Sean Lybrand, Sallie-Anne Pearson, Henry Cutler, Leonard Kritharides, Kathy Trieu, Quan Huynh, Tim Usherwood, Jo-An Occhipinti.

**Funding acquisition:** Jo-An Occhipinti.

**Methodology:** Cindy Q. Peng, Kenny D. Lawson, Mark Heffernan, Geoff McDonnell, Danny Liew, Sean Lybrand, Sallie-Anne Pearson, Henry Cutler, Leonard Kritharides, Kathy Trieu, Quan Huynh, Tim Usherwood, Jo-An Occhipinti.

**Project administration:** Cindy Q. Peng, Jo-An Occhipinti.

**Supervision:** Jo-An Occhipinti.

**Validation:** Mark Heffernan, Geoff McDonnell.

**Writing – original draft:** Cindy Q. Peng, Kenny D. Lawson.

**Writing – review & editing:** Cindy Q. Peng, Kenny D. Lawson, Mark Heffernan, Geoff McDonnell, Danny Liew, Sean Lybrand, Sallie-Anne Pearson, Henry Cutler, Leonard Kritharides, Kathy Trieu, Quan Huynh, Tim Usherwood, Jo-An Occhipinti.

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
