## [Decision Letter · Decision Letter 0]

2 Feb 2021

PONE-D-20-40440

Gazing through time and beyond the health sector: insights from a system dynamics model of cardiovascular disease in Australia

PLOS ONE

Dear Dr. Peng,

Thank you for submitting your manuscript to PLOS ONE. After careful consideration, we feel that it has merit but does not fully meet PLOS ONE’s publication criteria as it currently stands. Therefore, we invite you to submit a revised version of the manuscript that addresses the points raised during the review process.

Reviewer 2 in particular has pointed out that the presentation needs additional documentation in order to be consistent with best practice for publishing work in Systems Dynamics: both reviewers have suggested some more minor changes.

We look forward to receiving your revised manuscript.

Kind regards,

Susan Horton

Academic Editor

PLOS ONE

Journal Requirements:

2. In your Methods section, please provide additional information on the methods used, such as the equations representing the model; how the model was calibrated; and what parameters were applied.

"Mr. Lybrand reports salary and benefits from Amgen Australia Pty Ltd, the funder of this work.

A/Professor Atkinson reports a grant from Amgen Australia to undertake the research;  and A/Professor Jo-An Atkinson, Ms Cindy Peng, A/Professor Kenny Lawson, Professor Mark Heffernan, and Dr Geoff McDonnell report receiving funding from Amgen Australia to undertake the research.

A/Professor Jo-An Atkinson is Managing Director of Computer Simulation & Advanced Research Technologies (CSART), an international not-for-profit organization building infrastructure and capacity in the use of systems modelling and simulation to inform health and social policy."

Reviewers' comments:

Reviewer's Responses to Questions

**Comments to the Author**

1. Is the manuscript technically sound, and do the data support the conclusions?

Reviewer #1: Yes

Reviewer #2: Yes

2. Has the statistical analysis been performed appropriately and rigorously? 

Reviewer #1: Yes

Reviewer #2: Yes

3. Have the authors made all data underlying the findings in their manuscript fully available?

Reviewer #1: Yes

Reviewer #2: Yes

4. Is the manuscript presented in an intelligible fashion and written in standard English?

Reviewer #1: Yes

Reviewer #2: Yes

5. Review Comments to the Author

Reviewer #1: This is a very good article. There are only three omissions that need to be taken care of to make it an even better article. 1) You mention a group model building activity that provided the foundation of the model. It would be useful for the reader to know more about the insights that came out of that activity in the form of one or more causal diagrams that show some of the relationships that link interventions and outcomes. 2) The results shown in Table 2 require better explanation. For example, why do the results for the Medication and Treatment strategy look so much worse than the Supportive Environment strategy. To what extent is this because you have estimates for the interventions that make up the Medication and Treatment strategy and don't estimate costs for the Supportive environment? Why does the Pre-Hospital strategy produce a significant health system benefit? I would have expected better Pre-Hospital interventions would have saved lives of people who then require expensive hospital and follow up care including recurrent acute-CVD. If I've misunderstood, you need to explain better. In general, for results in both the graphs and tables, you need to relate the results better to the model's structure rather than simply reporting the results. 3) It would be interesting to see the impact of combined strategies and whether they can produce better results than single ones by complementing each others' strengths. This is often a lesson derived from SD health care and other models and a valuable lesson the those who might advocate for a particular class of interventions to the exclusion of others.

Reviewer #2: The article reports on outcomes of a participatory modeling projects to explore health policies that address the problem of cardiovascular diseases in Australia. Overall, the contribution of this work comes as a result of its participatory method (group model building) which brings many experts together and builds a simulation model that is consistent with the shared mental model of the participants. There is a rich history of group model building in system dynamics, and the method has proved to be effective to overcome modeling and implementation challenges.

Overall, I think this is a good work, and my suggestions are about improving the writings.

1- Please provide a better reference to the method and new developments in Group Model Building (GMB). GMB was introduced by George Richardson and David Andersen in their seminal works in 90s.

• Richardson, G. P., & Andersen, D. F. (1995). Teamwork in group model building. System Dynamics Review, 11(2), 113-137.

• Andersen, D. F., & Richardson, G. P. (1997). Scripts for group model building. System Dynamics Review: The Journal of the System Dynamics Society, 13(2), 107-129.

New advancements in the method including parameter estimation in group settings, model as a boundary object, and stakeholder analysis in GMBs are reported in these works:

• Luna‐Reyes, L. F., Black, L. J., Ran, W., Andersen, D. L., Jarman, H., Richardson, G. P., & Andersen, D. F. (2019). Modeling and simulation as boundary objects to facilitate interdisciplinary research. Systems Research and Behavioral Science, 36(4), 494-513.

• Hosseinichimeh, N., MacDonald, R., Hyder, A., Ebrahimvandi, A., Porter, L., Reno, R., ... & Andersen, D. F. (2017). Group model building techniques for rapid elicitation of parameter values, effect sizes, and data sources. System Dynamics Review, 33(1), 71-84.

2. Related to the first item, I strongly suggest the author to add a short reflection at the end to describes its methodological contributions, by reflecting on how the process worked, what they learned, and how they can further improve it, by potentially comparing and contrasting their works with the ones mentioned above. A paragraph in discussion would be enough.

3. Their model documentation is not meeting standards of a modeling work, and the need for reproducibility of scientific studies. To that end, first, the simulation model should be provided as a supplement. Second, the model equations (in mathemtaical format) should be all documented, so in case someone does not use Stella can replicate their findings in other softwares. Please refer to Rahmandad and Sterman’s article on minimum requirements for model documentation.

• Rahmandad, H., & Sterman, J. D. (2012). Reporting guidelines for simulation-based research in social sciences. System Dynamics Review 28(4): 396 – 411.

But I admit that the authors do a very good job of documenting parameter values and experimental designs in the Appendix.

4. I could not read the figures. The quality of the graphs, especially Figure 2, were very low. So my judgments are mainly based on the text. Please fix the figures.

5. Figure 1, the model: please add the inflow(s). (Looks to me the only inflow is birth rate and goes to No CVD). Further, guide the readers to go through the model by providing a descriptive text.

6. It appears that there are not many (any?) feedback loops in your model. This is fine as Homer and colleagues’ works of modeling cardiovascular disease also does not have many (any?) feedback loops. In limitations of the paper, please mention that the modelers focused on the compartmental modeling aspect of the project, and the project will benefit from exploring feedback mechanisms that can further affect the health statuses and demand for healthcare.

7. There is a history of health models in System Dynamics and specifically CVDs. Please check this source, as a very good literature review, and make sure major CVD modelings are not missed:

• Darabi, N., & Hosseinichimeh, N. (2020). System dynamics modeling in health and medicine: a systematic literature review. System Dynamics Review, 36(1), 29-73.

8. Minor: It might be better to spell out cardiovascular disease in the title, and maybe the first time it appears in the abstract, as CVD might not be a familiar term for many (I note that you mention it in the first page, which is good, but the title and abstract are unclear).

9. One of the findings of this work is that better screening, while helpful, leads to more demand for hospitalizations. That’s interesting (and you may talk more about it).

6. PLOS authors have the option to publish the peer review history of their article (what does this mean?). If published, this will include your full peer review and any attached files.

Reviewer #1: No

Reviewer #2: No

---

## [Author Response · Author response to Decision Letter 0]

1 Aug 2021

Reviewer #1: This is a very good article. There are only three omissions that need to be taken care of to make it an even better article. 

1) You mention a group model building activity that provided the foundation of the model. It would be useful for the reader to know more about the insights that came out of that activity in the form of one or more causal diagrams that show some of the relationships that link interventions and outcomes. 

Author response: Thanks. We have added a paragraph in page 19 discussing the learnings and our use of group model building. Since the goal was to build a simulation model and we anticipated the structure to be largely mechanistic based on similar CVD models, we used the participatory process to discuss mechanisms and focused on stock-and-flow structures rather than causal diagrams. The mechanisms and assumptions of interventions are described in details in the Appendix.

2) The results shown in Table 2 require better explanation. For example, why do the results for the Medication and Treatment strategy look so much worse than the Supportive Environment strategy. To what extent is this because you have estimates for the interventions that make up the Medication and Treatment strategy and don't estimate costs for the Supportive environment? Why does the Pre-Hospital strategy produce a significant health system benefit? I would have expected better Pre-Hospital interventions would have saved lives of people who then require expensive hospital and follow up care including recurrent acute-CVD. If I've misunderstood, you need to explain better. In general, for results in both the graphs and tables, you need to relate the results better to the model's structure rather than simply reporting the results. 

Author response: We appreciate Reviewer 1 pointing these out. We have added explanations in pages 15 and 17 to clarify. In general, the findings were described at a high-level rather than as explicit reference to model structure. For example, the reason for an increase in hospitalisations as a result of improvement in response to acute events was alluded to as survivors experiencing recurrent hospitalisations. 

3) It would be interesting to see the impact of combined strategies and whether they can produce better results than single ones by complementing each others' strengths. This is often a lesson derived from SD health care and other models and a valuable lesson the those who might advocate for a particular class of interventions to the exclusion of others.

Author response: Thank you, a valid point. We tested different combinations of strategies, which led to no significant findings. The model included a diverse range of interventions targeting different mechanisms, and it is largely compartmental in nature, with limited feedback loops. We agree that future modelling with a narrower scope and more depth should explore the interactions between different drivers and policies, and that synergistic effects would be important to note.

Reviewer #2: The article reports on outcomes of a participatory modeling projects to explore health policies that address the problem of cardiovascular diseases in Australia. Overall, the contribution of this work comes as a result of its participatory method (group model building) which brings many experts together and builds a simulation model that is consistent with the shared mental model of the participants. There is a rich history of group model building in system dynamics, and the method has proved to be effective to overcome modeling and implementation challenges.

Overall, I think this is a good work, and my suggestions are about improving the writings.

1- Please provide a better reference to the method and new developments in Group Model Building (GMB). GMB was introduced by George Richardson and David Andersen in their seminal works in 90s.

• Richardson, G. P., & Andersen, D. F. (1995). Teamwork in group model building. System Dynamics Review, 11(2), 113-137.

• Andersen, D. F., & Richardson, G. P. (1997). Scripts for group model building. System Dynamics Review: The Journal of the System Dynamics Society, 13(2), 107-129.

New advancements in the method including parameter estimation in group settings, model as a boundary object, and stakeholder analysis in GMBs are reported in these works:

• Luna‐Reyes, L. F., Black, L. J., Ran, W., Andersen, D. L., Jarman, H., Richardson, G. P., & Andersen, D. F. (2019). Modeling and simulation as boundary objects to facilitate interdisciplinary research. Systems Research and Behavioral Science, 36(4), 494-513.

• Hosseinichimeh, N., MacDonald, R., Hyder, A., Ebrahimvandi, A., Porter, L., Reno, R., ... & Andersen, D. F. (2017). Group model building techniques for rapid elicitation of parameter values, effect sizes, and data sources. System Dynamics Review, 33(1), 71-84.

Author response: Thank you. We have referenced the publication from Anderson and Richardson as the most relevant one to the process we have used (page 19).

2. Related to the first item, I strongly suggest the author to add a short reflection at the end to describes its methodological contributions, by reflecting on how the process worked, what they learned, and how they can further improve it, by potentially comparing and contrasting their works with the ones mentioned above. A paragraph in discussion would be enough.

Author response: We have added a paragraph on page 19 briefly discussing our process and how it compares to the group model building used in SD. Methodological differences with other non-SD modelling studies of CVD are discussed on pages 18 – 20.

3. Their model documentation is not meeting standards of a modeling work, and the need for reproducibility of scientific studies. To that end, first, the simulation model should be provided as a supplement. Second, the model equations (in mathemtaical format) should be all documented, so in case someone does not use Stella can replicate their findings in other softwares. Please refer to Rahmandad and Sterman’s article on minimum requirements for model documentation.

• Rahmandad, H., & Sterman, J. D. (2012). Reporting guidelines for simulation-based research in social sciences. System Dynamics Review 28(4): 396 – 411.

But I admit that the authors do a very good job of documenting parameter values and experimental designs in the Appendix.

Author response: We have added the full list of equations in the Appendix. We have also added that the simulation model can be provided on request. 

4. I could not read the figures. The quality of the graphs, especially Figure 2, were very low. So my judgments are mainly based on the text. Please fix the figures.

Author response: We have re-produced the figures for clarity.

5. Figure 1, the model: please add the inflow(s). (Looks to me the only inflow is birth rate and goes to No CVD). Further, guide the readers to go through the model by providing a descriptive text.

Author response: We have added an inflow of ‘turning 40 years of age’. A descriptive text of the model structure follows the reference to Figure 1 on page 9.

6. It appears that there are not many (any?) feedback loops in your model. This is fine as Homer and colleagues’ works of modeling cardiovascular disease also does not have many (any?) feedback loops. In limitations of the paper, please mention that the modelers focused on the compartmental modeling aspect of the project, and the project will benefit from exploring feedback mechanisms that can further affect the health statuses and demand for healthcare.

Author response: This is now covered by the new paragraph on page 19.

7. There is a history of health models in System Dynamics and specifically CVDs. Please check this source, as a very good literature review, and make sure major CVD modelings are not missed:

• Darabi, N., & Hosseinichimeh, N. (2020). System dynamics modeling in health and medicine: a systematic literature review. System Dynamics Review, 36(1), 29-73.

Author response: We have checked the CVD models cited in this systematic review and found no significant discrepancy. This systematic review has now been referenced on page 19.

8. Minor: It might be better to spell out cardiovascular disease in the title, and maybe the first time it appears in the abstract, as CVD might not be a familiar term for many (I note that you mention it in the first page, which is good, but the title and abstract are unclear).

Author response: This has been amended.

9. One of the findings of this work is that better screening, while helpful, leads to more demand for hospitalizations. That’s interesting (and you may talk more about it).

Author response: Screening leads to greater disease detection, and in some cases, higher demand for hospitalisations. However, we did not look at screening specifically. There is a finding that improvement in response to acute cardiovascular events leads to higher demand for hospitalisations, since the survivors can experience further hospitalisations. This is discussed on page 17.

---

## [Decision Letter · Decision Letter 1]

10 Sep 2021

Gazing through time and beyond the health sector: insights from a system dynamics model of cardiovascular disease in Australia

PONE-D-20-40440R1

Dear Dr. Peng,

We’re pleased to inform you that your manuscript has been judged scientifically suitable for publication and will be formally accepted for publication once it meets all outstanding technical requirements.

Kind regards,

Susan Horton

Academic Editor

PLOS ONE

Additional Editor Comments (optional):

Reviewers' comments:

Reviewer's Responses to Questions

**Comments to the Author**

1. If the authors have adequately addressed your comments raised in a previous round of review and you feel that this manuscript is now acceptable for publication, you may indicate that here to bypass the “Comments to the Author” section, enter your conflict of interest statement in the “Confidential to Editor” section, and submit your "Accept" recommendation.

Reviewer #1: All comments have been addressed

Reviewer #2: All comments have been addressed

2. Is the manuscript technically sound, and do the data support the conclusions?

Reviewer #1: Yes

Reviewer #2: Yes

3. Has the statistical analysis been performed appropriately and rigorously? 

Reviewer #1: Yes

Reviewer #2: Yes

4. Have the authors made all data underlying the findings in their manuscript fully available?

Reviewer #1: Yes

Reviewer #2: Yes

5. Is the manuscript presented in an intelligible fashion and written in standard English?

Reviewer #1: Yes

Reviewer #2: Yes

6. Review Comments to the Author

Reviewer #1: (No Response)

Reviewer #2: Thanks for addressing my comments. The manuscript looks good and ready for publication.

My only minor comment is that, on page 8, where you say "A SD model is a compartmental model primarily accounting for numbers of people in different states, called “stocks” (e.g. people living with CVD), and the numbers transitioning between the states, called “flows” (e.g. annual hospitalisations due to CVD),..." while widely accepted but it is challenged by many scholars of the field including John Sterman in his 2018 work of "System dynamics at sixty: the path forward." I personally agree with Sterman's point, as there are many good SD works that are not compartmental models. So to be more accurate, I would simply modify the beginning of your sentence to something like this: "Many conventional SD models include compartmental representations for number of people in different states, called “stocks” (e.g. people living with CVD), and the numbers transitioning between the states, called “flows” (e.g. annual hospitalisations due to CVD),..."

Otherwise the paper looks good, and ready for publication.

7. PLOS authors have the option to publish the peer review history of their article (what does this mean?). If published, this will include your full peer review and any attached files.

Reviewer #1: No

Reviewer #2: No

---

## [Editor Report · Acceptance letter]

22 Sep 2021

PONE-D-20-40440R1 

Gazing through time and beyond the health sector: insights from a system dynamics model of cardiovascular disease in Australia 

Dear Dr. Peng:

I'm pleased to inform you that your manuscript has been deemed suitable for publication in PLOS ONE. Congratulations! Your manuscript is now with our production department. 

Kind regards, 

on behalf of

Dr. Susan Horton 

Academic Editor

PLOS ONE